# Factors Affecting Continued Participation in Tuberculosis Contact Investigation in a Low-Income, High-Burden Setting

**DOI:** 10.3390/tropicalmed5030124

**Published:** 2020-07-29

**Authors:** Michelle May D. Goroh, Christel H.A. van den Boogaard, Mohd Yusof Ibrahim, Naing Oo Tha, Fredie Robinson, Khamisah Awang Lukman, Mohammad Saffree Jeffree, Timothy William, Anna P. Ralph

**Affiliations:** 1Department of Community and Family Medicine, Faculty of Medicine and Health Sciences, University Malaysia Sabah, Kota Kinabalu, Sabah 88400, Malaysia; michellegoroh@gmail.com (M.M.D.G.); naing_ot@ums.edu.my (N.O.T.); ssdrswe@ums.edu.my (S.); freddie@ums.edu.my (F.R.); khamisah@ums.edu.my (K.A.L.); saffree@ums.edu.my (M.S.J.); 2Menzies School of Health Research, Charles Darwin University, Darwin 0815, Australia; anna.ralph@menzies.edu.au; 3Borneo Medical and Health Research Centre, Faculty of Medicine and Health Sciences, University Malaysia Sabah, Kota Kinabalu, Sabah 88400, Malaysia; 4Infectious Disease Society Kota Kinabalu, Sabah 88200, Malaysia; tim7008@gmail.com

**Keywords:** tuberculosis contact screening, barriers, knowledge, perception, behavior, stigma

## Abstract

Setting: Outpatient clinics, Kota Kinabalu, Malaysia; January–April 2018. Objectives: To identify barriers to full participation in tuberculosis (TB) contact investigation. Methods: Cross-sectional study of knowledge, perceptions, and behaviours among TB contacts. This study was conducted among contacts who attended an initial clinic visit to explore retention in care. During this first visit, contacts were approached for participation in a questionnaire at a follow-up visit. Contacts who consented but did not subsequently attend were interviewed at home. Associations between questionnaire findings and attendance were tested using logistic regression. Results: Of the total 1436 identified contacts, 800 (56%) attended an initial clinic visit. Of 237 consenting TB contacts, 207 (87%) attended their follow-up appointment. In univariable analyses, the odds of attendance were highest for people notified to attend the TB clinic directly by a health inspector; close relatives of TB patients; non-students; people with higher incomes and smaller households; older individuals; males; and people not perceiving TB as stigmatising. In multivariable analysis, mode of notification to attend and having a close relative with TB remained significant. Conclusions: Health inspectors provide an effective role in TB contact investigation through direct personal communication to encourage the completion of the TB screening process, but this requires further integration with clinical processes, and with workplace and school-based investigations.

## 1. Introduction

Tuberculosis (TB) is the commonest infectious cause of death internationally. Globally, an estimated 10.0 million people fell ill with TB in 2018, which is a number that has been relatively stable in recent years. The burden of disease varies enormously among countries, from fewer than five to more than 500 new cases per 100,000 population per year, with the global average being around 130. Geographically, 44% of TB cases in 2018 were in the South-East Asia [1]. A key TB prevention activity is contact investigation and management for close contacts of people with infectious TB. This provides an opportunity to detect co-prevalent active TB among contacts and allows people with (latent) TB infection to be detected for monitoring or the commencement of treatment to prevent active disease [2]. WHO guidelines strongly recommend that after excluding active TB, high-risk contacts of confirmed pulmonary TB patients be commenced on preventive treatment. Those at highest risk are young children (aged < 5 years) and people of any age with immunosuppression [3]. Several effective preventive treatment regimens are recommended [3], the most commonly recommended being isoniazid as a daily oral medication for 6 months, which is known as isoniazid preventive therapy (IPT). The potential benefits are the greatest in young children because of their high risk of developing TB disease after exposure [4]. Contact investigation and management, incorporating TB preventive therapy plus the early detection of active disease, is a highly cost-effective strategy [5].

Tuberculosis (TB) is a disease of major public health significance in eastern Malaysia. While the national annual incidence of new TB patients is estimated at 80 per 100,000 population [6], Sabah state in the east, on the island of Borneo, has notification rates of 144 to 217 per 100,000 [7]. Key drivers of the TB epidemic in Sabah appear to relate to health-seeking behaviours [8].

Malaysian guidelines recommend that contact investigation using symptom screening, tuberculin skin test (TST) with or without a chest radiograph, and sputum collection should be undertaken for household members and other close contacts of people with TB [9]. The goal of the National TB Program is to identify and screen at least 10 contacts for every TB patient. Latent tuberculosis infection (LTBI) treatment is recommended and offered to those who are less than 5 years old or immunocompromised, but implementation has been limited. Health inspectors undertake contact investigation for each newly diagnosed TB patient, and all identified contacts are notified by health inspectors to participate in TB screening at the clinic, which requires attendance on several occasions. However, low participation in initial and follow-up appointments was identified during continuous quarterly ‘continuous quality improvement’ cycles held at the clinics as part of the broader health system-strengthening project for TB contact investigation, in which this study is embedded [10].

Barriers to participation in TB contact investigation have been investigated globally. Well-identified barriers at health system and patient levels contribute to the poor implementation of contact screening [11,12], including factors such as financial hardship, distance to clinics or cost of travel, time away from work, not understanding advice from healthcare workers, misunderstanding what TB is, or not understanding the benefit of screening. The specific barriers in Malaysia, a middle-income country with a high migrant population, have not been investigated. 

The aim of this research was to describe characteristics of TB contacts and identify barriers to full participation in TB contact investigation in Kota Kinabalu to inform locally relevant interventions. The study does not explore clinical outcomes of contact investigation. This study occurred within a broader health system-strengthening project to improve contact investigation policy implementation [10]. Specifically, we sought to explore the patient journey during the contact investigation process after initial contact with the clinic had been made. Barriers to retention in care need to be understood to ensure that system-strengthening strategies address appropriate factors.

## 2. Materials and Methods 

### 2.1. Design, Setting, and Explanation of Processes

This is a cross-sectional comparative study of TB contacts who attended or did not follow through with TB contact investigation after initially registering with a clinic. The study was undertaken at the two main TB clinics in Kota Kinabalu, the state capital of Sabah, Malaysia. This district has the highest TB burden in Sabah [7].

To identify contacts, a list with confirmed TB patients at the TB clinic is forwarded to health inspectors working for the government. Then, these health inspectors visit the homes of TB patients to list the patient’s contacts and details. During this visit, the identified contacts receive a letter with the request to attend the TB clinic for assessment within 2 weeks of notification. To notify workplace or school contacts, health inspectors communicate with workplace supervisors or class teachers/principals to identify individuals deemed at risk and inform them of their need for screening. These individuals receive the same letter with the request to attend for TB assessment. Contacts missed through these processes may self-present.

Payment for clinic attendance is waived for contacts presenting this personalised letter with the request for TB assessment, but self-presenters without a letter (approximately 15%) need to pay. Welfare is available only for Malaysian identity card holders if household income is <710 Malaysian Ringgit (RM) per month (USD 170). Contact investigation requires two or three clinic attendances (depending on onsite x-ray availability) for symptom review, X-ray, TST, and then follow up 3 days later to read the TST. One clinic has an onsite X-ray facility, but the other clinic does not.

### 2.2. Participants

Contacts of patients with newly diagnosed smear-positive pulmonary TB were documented according to the above procedures, with data entered into the TB registry (MyTB) and national ‘TBIS 10 C-2′ form by clinic staff. Contacts identified by health inspectors during the study period January to April 2018 who attended their first clinic appointment were eligible to participate in the study if they were 17 years or older and providing written, informed consent. Convenience sampling was used to invite contacts to participate in the study based on the availability of the research assistant who was employed by the project to enroll participants and undertake data collection. The research assistant was unavailable for each consecutive eligible contact, since she divided her time between both participating clinics to ensure consistency in the way interviews were conducted and interpreted. Contacts who did consent to participate but did not attend their follow-up visit within 2 weeks were categorised as non-attenders and interviewed at home shortly thereafter, when additional questions were asked as to investigate why they did not attend their appointment. Contacts seen at home had symptom-based screening and TST performed by a clinic nurse at home, a follow-up appointment 72 h later to read the TST result at home, and advice to follow-up additional screening components as necessary.

### 2.3. Data Collection

A questionnaire previously used in Vietnam to assess knowledge, attitudes, and practices among TB contacts [11] was used. Modifications were made for local relevance. Data included sex, ethnicity, education, employment, relationship to the TB index case, marital status, income, number of household members, and questions to assess knowledge, attitudes, and health-related behaviors (Appendix A). For people who did not attend their appointment, a question about why they did not attend was asked. Routinely collected data were accessed for information on outcomes of screening.

### 2.4. Definitions

A TB patient was defined as a symptomatic person with positive sputum microscopy. A TB contact was a person living in the same household or in frequent contact with the TB patient for at least 8 h a day [9]. Identification of contacts is made by Health Inspectors by visiting TB patients at home. Health inspectors employed through the Sabah State Health Department perform contact investigation for each newly diagnosed TB patient. All identified contacts are notified by health inspectors of the need to participate in TB screening at the clinics.

Malaysian nationality was determined on the basis of owning a Malaysian Identity Card. 

### 2.5. Analyses

The sample size calculation for a descriptive study of a dichotomous variable (attendance versus non-attendance at the clinic) was used [13]. We estimated that 85% of contacts would attend screening based on 2012–2016 data from the Sabah State Health Department TB/Leprosy Control Unit. To estimate proportions accurately at the 95% confidence level (alpha 0.05), a sample size of 196 was calculated. For an estimated non-respondent rate of 20%, 235 people would need to be approached.

Analyses were undertaken in SPSS Statistics v22 and Stata version 15.2, and figures were made in Microsoft Excel v10. Chi-squared or Fisher’s exact tests were used to compare proportions, and the Wilcoxon rank-sum test was used to compare non-normally distributed continuous variables. Univariable and multivariable logistic regression models were used to identify the odds of attendance according to questionnaire responses. Variables included in the survey that had a plausible association with the outcome were all included in the univariable analyses. The multivariable model was constructed by including variables that had a plausible association with the outcome and were statistically associated with the outcome in univariable analyses at *p* < 0.05. 

### 2.6. Ethics

Written ethical approval for this study was granted by the Malaysian Medical Research and Ethics Committee (NMRR-17-2139-37893 (IIR)) and the University of Malaysia Sabah (JKEtika 1/18 (11)).

## 3. Results

During January–April 2018, 202 TB patients were registered at the participating clinics and 1436 contacts were identified (Figure 1). Of those contacts, 672 (47%) were female and 800 (56%) did attend the clinic for the first screening appointment. Study staff approached 339 (42.4%) contacts, and 237 (69.9%) consented and were interviewed. Illustrative of the challenges of delivering TB care in this environment, a fire at one participating site contributed to not all contacts being approached, as clinic operations scaled down and a number of TB patients and their contacts were required to receive care at non-study sites during clinic re-construction. Overall, 207 (87.3%) attended the clinic for their scheduled follow-up appointment within 2 weeks; 30 (12.7%) did not.

The majority of participants were female (67.9%), median age 28 (range 17–66); a quarter were non-Malaysian (Table 1). Date of birth was missing for people without an identity card (non-Malaysians). The median number of household members was 6 (range 1–15). Twenty-three percent had no formal education. Most (85.6%) were of low-income status with monthly income <2000 RM (approximately US $480) per month. Income was associated with ethnicity (higher in Malaysians than non-Malaysians) and education. Most contacts (201/237 [84.4%]) were advised directly to participate in screening. The rest were advised via their school or workplace. Most (198/237 [83.5%]) lived <10 km from the clinic.

### 3.1. Knowledge, Attitudes and Practices

The majority recognised that TB was caused by an infection (97.9%) and that one could acquire TB through sleeping close to someone with active TB (78.9%) (Table 2). Participants had uncertainty about TB symptoms, with only 65.8% nominating correctly that fever could be a feature of TB (Table 2). Many thought TB could be transmitted by sharing utensils or belongings and were unaware of non-pulmonary forms of TB. Sixteen percent believed traditional medicine could cure TB. 24/75 [32.0%] people with limited education versus 14/162 [8.6%] people with higher educational levels [*p* <0.001] (Figure 2). Most (234/237 [98.7%]) thought TB was curable by taking proper treatment. 

Fifty-three people (22.4%) thought there was discrimination; 16/30 (53.3%) of non-attenders and 37/207 (17.9%) of attenders (*p* <0.001, Figure 3). There was an association between the perception of stigma and being a close relative of a TB index case. All but one agreed that household TB screening was worthwhile. Most (87.8%) recognised they were at elevated risk of TB. However, nearly a quarter indicated that they would wait 3–4 weeks before going to the clinic if they had TB symptoms (Table 3).

### 3.2. Associations with Attendance

In univariable analyses (Table 4), the strongest associations with attendance were the relationship of the contact to a TB case, student status, and how notification was received. The odds of attendance for close relatives versus others was 101.5 (95% CI 13.5–765.3); for non-students, it was 8.5 (95% CI 3.7–19.4); for people who received direct notification, it was 128.1 (37.5–437.9). There was a roughly inverted U-shaped curve in the relationship between income and attendance (Table 1). When this was summarised as a dichotomous variable, those with very low income (< 1000 RM) were less likely to attend (Table 4).

In multivariable analyses, the strategy by which people were notified to attend, and whether the individual was a close relative of an index case, were significantly associated with attendance. These variables were related to each other: 26/30 of non-attenders were people who did not have a close relative with TB and had been informed of the need for contact investigation through their workplace or school. Age and perception of stigma could not be included in the multivariable model due to co-linearity (age not documented for non-Malaysians and high numbers of ‘don’t know’ responses to the question on stigma among non-Malaysians). 

### 3.3. Interviews with Non-Attenders

The main reason to not participate in contact investigation was difficulty getting time off work or school (27/30 people, Table 3). Two indicated they were concerned about discrimination, and one indicated that distance to the clinic was prohibitive.

## 4. Discussion

In this high TB-burden setting, the majority (87.3%) of TB contacts attended their follow-up visit for screening within two weeks of initially making contact with the clinic. Failure to attend subsequent scheduled appointments was more common if information about the need for contact screening was provided ‘passively’ through workplaces or schools, and if contacts were not close relatives of someone with active TB. Notification through workplaces or schools is likely to be perceived as less urgent than a home visit from a health inspector. The experience of having a household member with active TB could be a motivating factor for participation in contact investigation. The availability of health inspectors to conduct household visits appears to be a highly valuable resource. Creating a more active role for them in workplace and school-based screening will be important in addressing reasons for non-attendance identified here. This study was conducted among the 56% of contacts who made initial contact with the clinic and explores their subsequent journey through the contact investigation process to explore retention in this process. Findings are not necessarily representative of all TB contacts; the remaining 44% who made no contact with the clinic at all may experience different barriers. However, these remaining contacts are subsequently followed up for the next scheduled appointments for screening, which is at the third month, ninth month, and 21st month from the date of contact identification.

An additional factor associated with non-participation in screening appeared to be fear of discrimination. Although a minority of people cited concern about stigma, it was the barrier articulated by two non-attenders (53.3% non-attenders were concerned about stigma compared with 17.9% attenders). The perception of stigma being less among relatives of TB patients is reassuring. This suggests that knowing someone with the condition is destigmatising and may reflect participants’ observations of how their TB-affected family member is treated. 

Our study in a middle-income country found an inverted U-shaped relationship between clinic attendance and income; this may be attributable to difficulties paying for transport for low-income individuals and difficulty leaving the workplace for high-income individuals. Studies from globally diverse settings have identified cost as a critical barrier to TB contact investigation [11] or TB care [14]. The End TB strategy articulates the elimination of catastrophic costs due to TB as the earliest priority, proposing that by 2020, 0% of families should be facing catastrophic costs. ‘Catastrophic’ is defined as total costs of TB care exceeding 20% of annual household income [15]. For people in our study, costs for 1–3 days away from work and transportation would be well below this definition. However, costs could still pose a high enough disincentive to deter attendance. Especially since contacts are largely asymptomatic, ‘willingness to pay’ for contact investigation is likely much lower than willingness to pay for the treatment of active TB. The investigation and management of close TB contacts has been shown to be highly cost effective [5], but for contacts, numerous days away from income-generating activities, the need to attend the clinic in person, and in some cases to pay for clinic attendance are all the main financial barriers for contacts to attend screening [14] and diminish total cost effectiveness. Reducing the onerousness of contact tracing in our study setting such as using a ‘one-stop shop’ approach, removing the need for TST over at least 2 clinic visits by implementing the WHO recommended symptom-based screening approach during one visit [3], extending clinic opening hours, and/or the provision of outreach services could all help improve the uptake and cost-effectiveness [5] of contact investigation. 

State-supported financial welfare is unavailable to foreigners who comprise a significant proportion of TB patients and contacts, and the lowest-income bands. Health providers should seek alternative sources of financial support for them such as non-governmental organisations and foreign embassies. A recent Fee Act was enforced by the Malaysian Ministry of Health, requiring that patients attending government health facilities pay significant fees if they lack a Malaysian identity card. To improve equity, revisions should be made to the current health system by the Ministry of Health in collaboration with involved stakeholders. Supporting the most vulnerable members of society in accessing health care will be beneficial for society overall. 

This study showed reasonable knowledge of TB. Choices about health-seeking behaviour were generally appropriate. This contrasts with a Tanzanian study that found limited community TB knowledge [16]. This suggests that health promotion activities in Kota Kinabalu have some effectiveness; these must be continued and strengthened to better reach those most at risk, especially migrants. In a Thai study, distance to clinics was a factor in non-attendance [17]; while this was not significant in our study, one non-attender did cite this as a disincentive (Table 3). The proportion of people attending for screening in this study was similar to a study from Vietnam [11]. However, associations with attendance differed substantially between these studies, with incorrect beliefs about TB being associated with non-attendance in the Vietnamese study. These findings highlight the importance of local knowledge to inform locally relevant responses. 

A study of TB case finding from eastern Sabah concluded that fear of TB among immigrant workers could lead to avoidance of seeking healthcare [8]. Compared to legal migrants in Malaysia, undocumented migrants face greater health and socioeconomic inequalities and confront higher structural barriers. Findings in this study confirm the over-representation of non-Malaysians among TB contacts in general (26.6%), and among disadvantaged strata; however, their attendance rates for contact investigation was high (96.8%), suggesting a strong desire to engage. It is unclear what proportion of those who never attended the clinic (and hence were not enrolled in the study) may have been migrants. 

A limitation of the study is that people who consented differed from the overall cohort, being more likely to be female, and to attend the clinic than the total 1436 contacts identified during the study period. Therefore, we acknowledge that the collated data are more representative of female attenders than of the overall cohort of contacts. This study was conducted among the 56% of contacts who attended their first appointment in the clinic and may not be representative of the 44% of contacts not attending the clinic initially. For further improvement of the contact investigation system, it is important to investigate in future research if the barriers for contacts not attending their initial appointment are similar to contacts attending their first appointment, to ensure that system-strengthening strategies address appropriate factors for all contacts. Of the people attending their first appointment in the clinic, 42% were invited to participate through convenience sampling rather than a random selection. This sampling was dependent on the study nurse being present at the clinic. We do not expect that this would have introduced selection bias in favour of specific characteristics of participants. However, the 30% who did not consent to participate (Figure 1) might have introduced selection bias towards people who were more likely to participate in general, including contact screening.

Another limitation of this study is potential response bias. Participants may have provided most palatable responses: for example, citing an inability to have time off work may be easier than saying they distrusted the clinic. However, efforts were made to ensure that participants were treated respectfully to encourage honest answers, and participants were aware that research staff administering the questionnaire were independent from clinic staff. 

## 5. Conclusions

Health systems need to respond innovatively to community needs to implement stronger TB control. This study shows the value of existing resources such as health inspectors in promoting TB contact investigation, but this requires closer integration with clinical processes and a greater role in workplace and school-based investigations. Improved public awareness of TB focusing on early health seeking and stigma reduction remain important priorities. The feasibility of extended clinic hours and home visits need to be investigated. A health system-strengthening project is in progress at the participating clinics to implement such measures.

## Figures and Tables

**Figure 1 tropicalmed-05-00124-f001:**
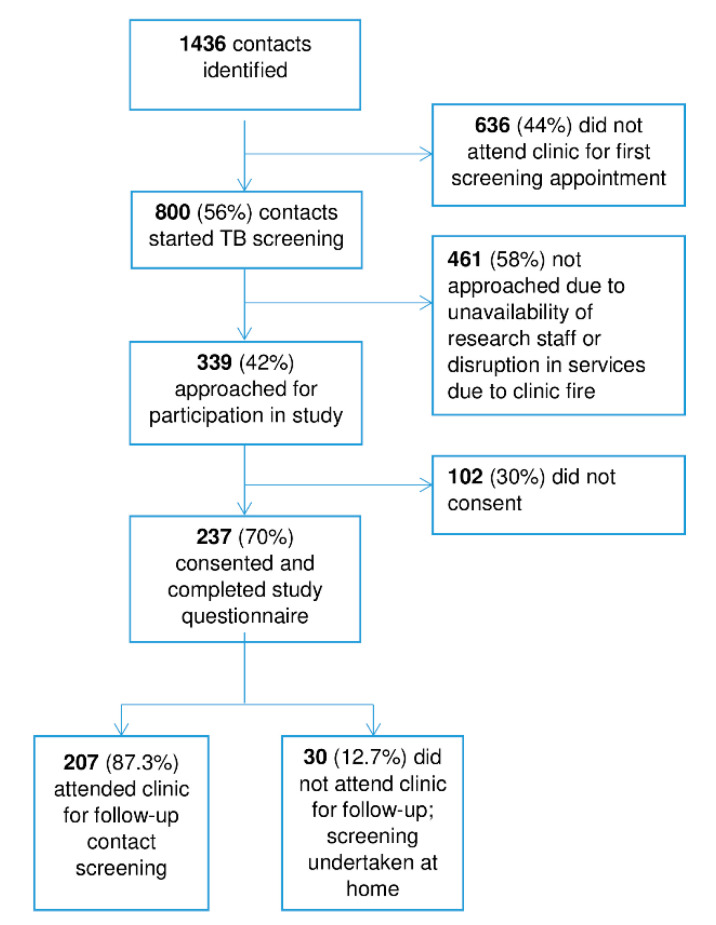
Study diagram.

**Figure 2 tropicalmed-05-00124-f002:**
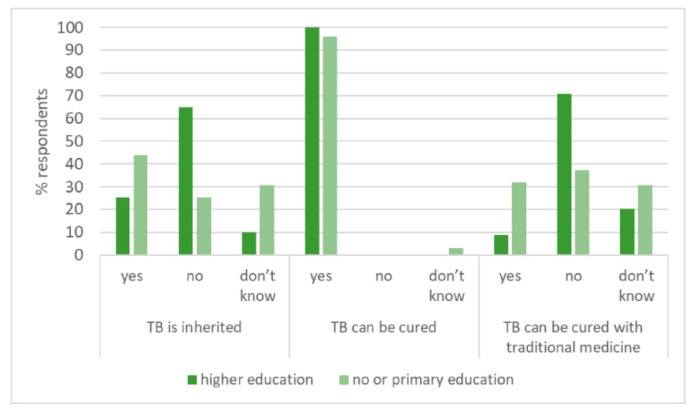
Associations between selected TB knowledge questions and educational status.

**Figure 3 tropicalmed-05-00124-f003:**
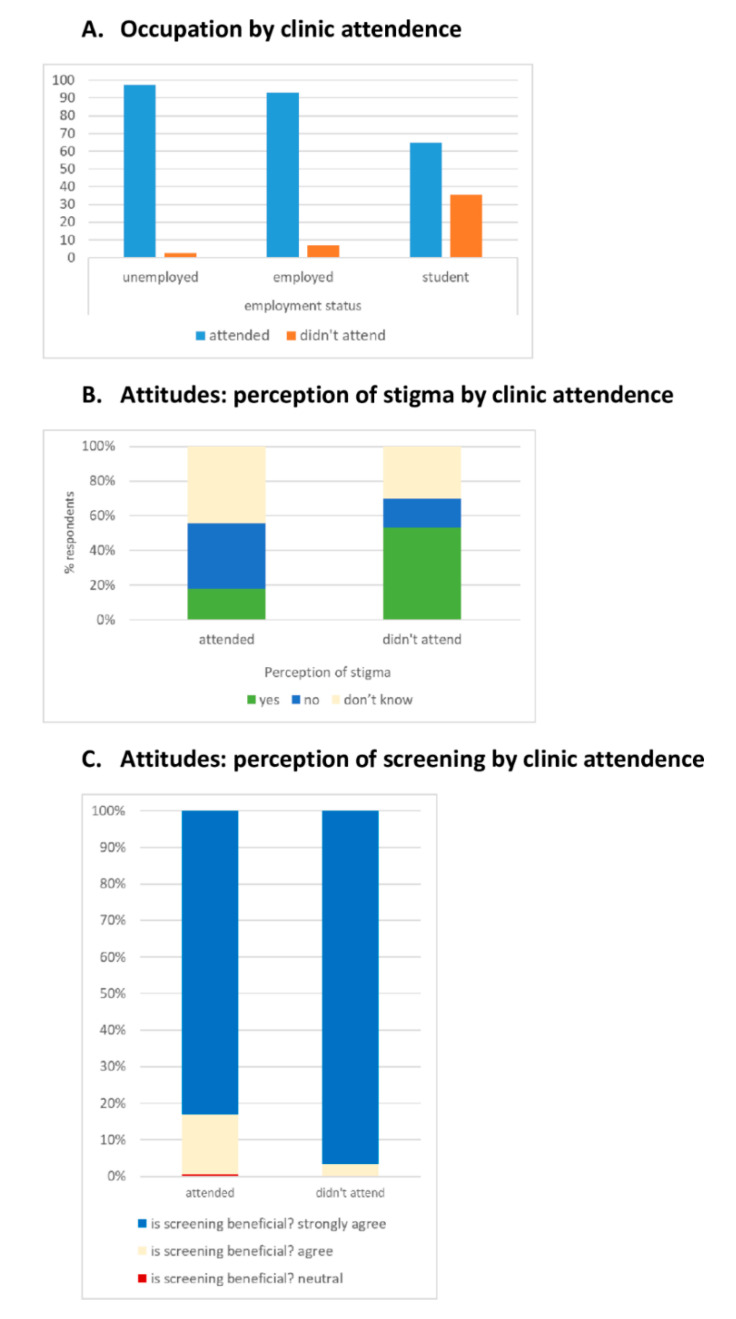
Association between characteristics or beliefs and attendance at clinic for contact screening Study diagram.

**Table 1 tropicalmed-05-00124-t001:** Demographic characteristics among attending and non-attending study contacts. TB: tuberculosis.

Characteristics	All	Attended	Did not Attend	*p*
**Number**	237	207 (87.3%)	30 (12.7%)	
**Age (median, range)**	28 (17–66)	30.5 (17–66)	17 (17–53)	0.001
**Sex**				
Male: number (%)	76 (32.1)	72 (94.7%)	4 (5.3%)	
Female: number (%)	161 (67.9%)	135 (83.9%)	26 (16.2%)	0.021
**Nationality**				
Malaysian	174 (73.4)	146 (83.9%)	28 (16.1%)	
Non-Malaysian	63 (26.6)	61 (96.8%)	2 (3.2%)	0.007
**Ethnicity**				
Sabah Indigenous (Bumiputera)	140 (59.1)	116 (82.9%)	24 (17.1%)	
Malay	4 (1.7)	3 (75.0%)	1 (25.0%)	
Chinese Malaysian	16 (6.8)	15 (93.8%)	1 (6.3%)	
Indian Malaysian	1 (0.4)	1 (100.0%)	0 (0.0%)	
Migrants	76 (32.1)	72 (94.7%)	4 (5.3%)	0.11
**Level of education**				
Bachelors’ and above	7 (3.0)	4 (57.4%)	3 (42.9%)	
Diploma	24 (10.1)	18 (75.0%)	6 (25.0%)	
High school	25 (10.5)	25 (100.0%)	0 (0.0%)	
Middle school	106 (44.8)	87 (82.1%)	19 (17.9%)	
Primary school	20 (8.4)	20 (100.0%)	0 (0.0%)	
No formal education	55 (23.2)	53 (96.4%)	2 (3.6%)	<0.001
**Current main job**				
Unemployed	9 (3.8)	9 (100.0%)	0 (0.0%)	
Housewife	29 (12.2)	28 (96.6%)	1 (3.5%)	
Self-employed	72 (30.4)	72 (100.0%)	0 (0.0%)	
Private employee	50 (21.1)	45 (90.0%)	5 (10.0%)	
Government employee	20 (8.4)	16 (80.0%)	4 (20.0%)	
Student	54 (22.8)	35 (64.8%)	19 (35.2%)	
Other	3 (1.3)	2 (66.7%)	1 (33.3%)	<0.001
**Relationship to TB index case**				
Parent	42 (17.7)	42 (100.0%)	0 (0.0%)	
Spouse	40 (16.9)	39 (97.5%)	1 (2.5%)	
Child	15 (6.3)	15 (100.0%)	0 (0.0%)	
Sibling	20 (8.4)	20 (100.0%)	0 (0.0%)	
Other relative	45 (19.0)	45 (100.0%)	0 (0.0%)	
Other	75 (31.6)	46 (61.3%)	29 (36.7%)	<0.001
**Marital status**				
Married	118 (49.8)	111 (94.1%)	7 (5.9%)	
Single	119 (50.2)	96 (807%)	23 (19.3%)	0.003
**Monthly income (Malaysian Ringgit, RM *)**				
<RM 710	65 (27.4)	46 (70.7%)	19 (29.2%)	
RM 710–RM 1000	61 (25.7)	59 (96.7%)	2 (3.3%)	
RM 1001–RM 2000	77 (32.5)	76 (98.7%)	1 (1.3%)	
RM 2001–RM 3000	16 (6.8)	16 (100.0%)	0 (0.0%)	
>RM 3000	18 (7.6)	10 (55.6%)	8 (44.4%)	<0.001
Number of contacts in household: median (range)	6 (1–15)	5 (2–15)	6.5 (1–12)	0.02

* Malaysian Ringgit (RM) approximately equal to 20 US cents. <1000 RM/month is considered to be very low income, <2000 RM/month low income and >2000 RM/month high income.

**Table 2 tropicalmed-05-00124-t002:** Knowledge among contacts of tuberculosis patients.

Characteristics (n = 237)	Frequency (%)		
**TB is commonly associated with the following symptoms:**	**Yes**	**No**	**Don’t know**
Cough	232 (97.9)	3 (1.3)	2 (0.8)
Leg pain	91 (38.4)	86 (36.3)	60 (25.3)
Weight loss	194 (81.9)	21 (8.9)	22 (9.3)
Night sweats	160 (67.5)	30 (12.7)	47 (19.8)
Coughing up blood	231 (97.5)	4 (1.7)	2 (0.8)
Fever	156 (65.8)	38 (16.0)	43 (18.2)
**TB can be transmitted by:**			
Sneezing	214 (90.3)	15 (6.3)	8 (3.4)
Sharing utensils	182 (76.8)	41 (17.3)	14 (5.9)
Sleeping in the same room	187 (78.9)	38 (16.0)	12 (5.1)
Sharing a toilet	139 (58.6)	70 (29.5)	28 (11.8)
Sexual intercourse	138 (58.2)	65 (27.4)	34 (14.3)
**TB is caused by:**	**True**	**False**	**Don’t Know**
An infection	232 (97.9)	2 (0.8)	3 (1.3)
Unhygienic environment	228 (96.2)	3 (1.3)	6 (2.5)
Inherited from parents	74 (31.2)	124 (52.3)	39 (16.5)
**The following people have higher TB risk:**			
Smokers	213 (81.9)	14 (5.9)	10 (4.2)
Children	194 (81.9)	12 (5.1)	31 (13.1)
People with HIV/AIDS	208 (87.8)	4 (1.7)	25 (10.5)
Pregnant women	154 (65.0)	23 (9.7)	60 (25.3)
**Treatment and cure of TB:**			
TB can be completely cured if a patient takes treatment.	234 (98.7)	0 (0.0)	3 (1.3)
Traditional medicine can be used to cure TB	38 (16.0)	143 (60.3)	56 (23.7)

**Table 3 tropicalmed-05-00124-t003:** Perceptions and practices in the whole cohort, and the subgroup of non-attenders.

Question	Frequency (%)
**Number**	**237**
**Perceive that TB is stigmatised (‘Do you think that there is discrimination against people with TB?’)**	
Yes	53 (22.4)
No	83 (35.0)
Don’t know	101 (42.6)
**Believe that own risk of TB is higher than the general population**	
Much lower risk	2 (0.8)
Lower risk	5 (2.1)
Similar risk	22 (9.3)
Higher risk	79 (33.3)
Much higher risk	129 (54.5)
**Believe TB screening is beneficial for their family**	
Neither agree nor disagree	1 (0.4)
Agree	35 (14.8)
Strongly agree	201 (84.8)
**What would you do if you had symptoms of TB?**	
Go to health facility	232 (97.9)
Go to pharmacy	4 (1.7)
Go to traditional healer	1 (0.4)
**When would you to go to a health facility if you had TB symptoms?**	
When symptoms that look like TB signs last for 3–4 weeks	56 (23.6)
As soon as I realise that my symptoms might be related to TB	181 (76.4)
**Number**	**30**
**I didn’t attend the clinic appointment because:**	
The distance to travel from my home to preventative district health centre is too far	1 (3.3)
I am worried about discrimination from other people toward myself and my family	2 (6.7)
It is time-consuming. It was difficult to get time off work or study	27 (90.0)

**Table 4 tropicalmed-05-00124-t004:** Factors associated with clinic attendance for contact investigation.

	Univariable			Multivariable		
	Odds Ratio *	95% CI	*p* Value	Odds Ratio *	95% CI	*p* Value
Age	0.19	1.03–1.15	**0.001**	**	-	-
Male (1: male, 0: female)	3.47	1.16–10.31	**0.026**	0.49	0.06–3.93	0.505
Malaysian (1: Malaysian, 0: non-Malaysian)	0.17	0.04–0.74	**0.018**	0.74	0.26–21.35	0.863
Low educational status (1: low education, 0: higher education)	7.63	1.77–32.93	**0.006**	0.35	0.01–11.95	0.559
Student (1: non-student, 0: student)	8.49	3.71–19.4	**<0.001**	0.40	0.01–19.37	0.640
Close relative (1: close relative, 0: other)	101.5	13.5–765.3	**<0.001**	41.77	3.28–531.22	**0.004**
Married (1: married, 0: single)	3.80	1.56–9.24	**0.003**	0.43	0.59–1.09	0.396
Income status (1: income>1000 RM, 0: income <1000 RM)	2.36	1.03–5.38	**0.042**	1.82	0.06–51.4	0.729
Household occupancy (continuous variable between 1 and 15)	0.83	0.70–0.97	**0.020**	0.81	0.60–1.09	0.159
Perception of stigma (1: yes, 0: no)	0.15	0.05–0.44	**0.001**	**	-	-
Whether the respondent would delay clinic attendance if TB symptoms were present (1: go quickly, 0: delay 3–4 weeks)	0.46	0.15–1.38	0.164	-	-	-
Distance to clinic (1: 0-9km, 0: ≥10km)	0.33	0.07–1.44	0.140	-	-	-
How the respondent was informed (1: letter, phone, visit, 0: notification to work/school)	128.05	37.45–437.89	**<0.001**	89.3	11.1–149.10	**<0.001**

* Odds ratio of attending. ** Age and stigma unable to be included in multivariable model due to collinearity.

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
