# Peer review of "Factors Affecting Continued Participation in Tuberculosis Contact Investigation in a Low-Income, High-Burden Setting"

_tropicalmed, 2020, doi:10.3390/tropicalmed5030124_

Round 1

Reviewer 1 Report

General comments

This study was conducted among contacts of active TB cases in Malaysia. The study instruments and analyses are appropriate. A major limitation of the study is that the findings a not likely representative of all contacts due to the selection process.

Specific comments

Abstract

P1 line 21 – Please add that the study was conducted among the 56% of contacts who attended their first clinic visit.

Introduction

P2 line 49- It is not clear what +/- means. Does this mean “as clinically indicated?” When is a chest radiograph recommended and/or done? Is treatment of LTBI recommended and provided? Describe the goals of the program then note that this study is only evaluating one of the initial steps in contact investigation, namely attending the initial clinic appointment, but not TST placement, interpretation or additional steps in the process.

P2 line 51- Briefly note who and how identification of contacts is done. Is this done by health inspectors? This can also be done in the Definitions page 3.

Methods

P2 line 85 – Please indicate how “a subset of all contacts” was selected. Was this a convenience sample or a randomly selected group meant to be representative of all contacts who attended their first clinic visit?

Results

P3 lines 121-122- and Figure 1- The data here are not consistent with the high expected attendance of 85% with 44% not attending the first visit and then excluded from being selected for the study. Is it correct that study participants included only those who attended their appointment? This suggests the study was conducted about the reasons for not attending a scheduled follow-up visit, but only after attending the initial visit.

Discussion

P10 line 193- Please add that this study was conducted among the 56% of contacts who attended their first clinic visit, and it may not be representative of all contacts.

P10 line 224- What does “removing the need for a TST” mean?

P11 line 254- Please add that a major limitation is that the study was done among the 56% who attended the first appointment and may not be representative of the 44% not attending. If an attempt was not made for a random selection of those invited to participate, this would be another limitation.

Author Response

We would like to thank the reviewers for their thoughtful review and feedback of our manuscript with the title “Factors affecting continued participation in tuberculosis contact investigation in a low-income, high-burden setting” submitted for publication in the Tropical Medicine and Infectious Diseases special issue: “Tuberculosis Elimination in the Asia-Pacific”.

We responded to the reviewers queries and updated the manuscript accordingly. Please find attached the relevant documents including:

  • Revised manuscript with track changes
  • Revised manuscript - clean version
  • Responses to feedback Reviewer 1
  • Responses to feedback Reviewer 2

Thank you for your time in reviewing this manuscript and we look forward to your feedback. Please let us know if there is any additional information that you require.

Yours sincerely,            

Dr Michelle Goroh           &             Prof Anna Ralph

Reviewer 2 Report

This paper highlights the importance of contact investigation which is very much needed. 

Over all comment:

Please use WHO approved Standardized terms for TB patients. Please do not use the term “cases”- it is stigmatizing. Rather use “TB patients” if they are already diagnosed. Or people with possible TB if they have not been diagnosed as yet. 

Background - needs strengthening

Global data needs to be presented better. 

There seems to be an emphasis on contact investigation amongst migrant population. But then the objective does not explicitly state that. Is this paper focusing on contact investigation in general or only for migrant population. 

Methods

How were a subset of patients were chosen for contact investigation 

I think its important to clearly identify the 2 sub sets of patients in the methods. 

Def of contact: what is frequent contact ?- it will be helpful to have precise definitions.

After what duration participants were visited at home 

The authors mention that one nurse was recruited by the study. What was her job specifically. Are the other personnel already working for the TB program. Does the program have dedicated contact investigation personnel. 

The authors mention in passing about a fire at the clinic. What was its impact -- must be clearly stated. 

Contact investigation is part of any good TB control program. So how was the enrollment different in this study as over 58% of people could not be enrolled due to staff shortage. This goes to the question above of dedicated contact investigation staff. A good patient flow diagram will help. 

Results:

Mostly it is descriptive analysis except for table 4 where they look at factors associated with attendance. What is not clear is how the authors picked the variables for the univariate analysis. Were these everything they included in their survey? Also, the variable selection procedure using significant testing is suspect. They can use a different selection procedure or include all variables in the model unless there are collinearity issues. If pre-screening of variables needs to be done then they should choose a more liberal cutoff of 0.2.

A good reference on variable selection: https://onlinelibrary.wiley.com/doi/pdf/10.1111/tri.12895

As the authors have focused on migrant population - is there a merit in presenting these data separately to assess if they are at a disadvantage. 

Discussion

Cost effective statements and factors that "diminish cost effectiveness" needs reference. 

A big limitation of the study is its enrollment of 42% only. 

Author Response

(The authors gave the same response as above.)

Round 2

Reviewer 1 Report

The manuscript now clearly states the analysis is based on the 56% of registered contacts and the implications. This should be noted in the abstract as well. If one believes evaluation of contacts is important, the missing group should not be overlooked. 

Why not add a sentence or two in the discussion about the missing 44%? These individuals are not screened at all, correct? Is this important? How could this be improved upon or at least the reasons investigated?

Author Response

Thank you for your time and insights in preparing the review, 

Please see attachment for our response.

With kind regards, 

Reviewer 2 Report

no further comments. 

Author Response

Thank you.

With kind regards